# Combining stochastic resetting with Metadynamics to speed-up molecular dynamics simulations

Ofir Blumer[1], Shlomi Reuveni [1,2,3] & Barak Hirshberg [1,2,3] ✉

Metadynamics is a powerful method to accelerate molecular dynamics simulations, but its efficiency critically depends on the identification of collective variables that capture the slow modes of the process. Unfortunately, collective variables are usually not known a priori and finding them can be very challenging. We recently presented a collective variables-free approach to enhanced sampling using stochastic resetting. Here, we combine the two methods, showing that it can lead to greater acceleration than either of them separately. We also demonstrate that resetting Metadynamics simulations performed with suboptimal collective variables can lead to speedups comparable with those obtained with optimal collective variables. Therefore, applying stochastic resetting can be an alternative to the challenging task of improving suboptimal collective variables, at almost no additional computational cost. Finally, we propose a method to extract unbiased first-passage times from Metadynamics simulations with resetting, resulting in an improved tradeoff between speedup and accuracy. This work enables combining stochastic resetting with other enhanced sampling methods to accelerate a broad range of molecular simulations.

Molecular dynamics (MD) simulations provide valuable insights into the dynamics of complex chemical and physical systems. They are a powerful tool but, due to their atomic spatial and temporal resolution, they cannot be applied to processes that are longer than a few microseconds, such as protein folding and crystal nucleation[1–3]. Different methods have been developed in order to overcome this timescale problem, such as umbrella sampling[4,5], replica-exchange[6], free-energy dynamics[7,8], milestoning[9,10], weighted ensemble[11–13], Metadynamics (MetaD)[14–18], On-the-fly probability enhanced sampling (OPES)[19–22], and many others. In this paper, we will focus on MetaD, which relies on identifying efficient collective variables (CVs), capturing the slow modes of the process, and introducing an external bias potential to enhance the sampling of phase space along them. The ability of MetaD to accelerate simulations crucially depends on the quality of the CVs[18,23,24]. An optimal CV is capable of distinguishing between metastable states of interest as well as describing their

interconversion dynamics[25,26]. A suboptimal CV can lead to hysteresis, and poor inference of the unbiased free-energy surface or kinetics[1,14,23,27,28].

Very recently, we developed a CV-free approach for enhanced sampling, based on stochastic resetting (SR)[29]. Resetting is the procedure of stopping stochastic processes, at random or fixed time intervals, and restarting them using independent and identically distributed initial conditions. It has received much attention recently[30,31], since it is able to expedite various processes ranging from randomized computer algorithms[32–34] and service in queuing systems[35], to first-passage and search processes[36–46]. We demonstrated the power of SR in enhanced sampling of MD simulations, showing it can lead to speedups of up to an order of magnitude in simple model systems and a molecular system[29]. Moreover, we developed a method to infer the unbiased kinetics, in the absence of SR, from simulations with SR.

[1]School of Chemistry, Tel Aviv University, Tel Aviv 6997801, Israel. [2]The Center for Computational Molecular and Materials Science, Tel Aviv University, Tel Aviv 6997801, Israel. [3]The Center for Physics and Chemistry of Living Systems, Tel Aviv University, Tel Aviv 6997801, Israel. ✉e-mail: hirshb@tauex.tau.ac.il

Resetting is an appealing method due to its extreme simplicity: it can be trivially implemented in MD codes, and no CVs are required. Since finding good CVs in complex condensed phase systems is a difficult challenge, it is a potentially significant advantage. Furthermore, unlike other methods, which continuously add energy to the system, SR does not change the dynamics between resetting events. However, acceleration by SR is not guaranteed. A sufficient condition is that the distribution of transition times (also called first-passage times, FPTs) of the corresponding process is wide enough, i.e., it has a standard deviation that is greater than its mean[47]. Many systems, including the models and molecular example discussed in our previous work[29], fulfill this criterion but there are also counter examples.

The complementary advantages and limitations of MetaD and SR raise an important question: can we combine them to obtain the best of both worlds? Since SR can be applied to any random process, we can restart MetaD simulations as previously done for unbiased simulations. This observation opens the door for using SR as a complementary tool to existing enhanced sampling procedures.

In this paper, we combine SR with MetaD. In one model system, we show that this approach leads to greater acceleration than any of them independently, even in comparison to using the optimal CV in MetaD simulations. In another model system, we restart MetaD simulations, performed with suboptimal CVs, and obtain accelerations similar to those achieved using the optimal CV. This result suggests that a straightforward application of SR can be an alternative to the challenging task of improving a suboptimal CV. We then demonstrate this for transitions between metastable states of alanine tetrapeptide, and for the folding of the mini-protein chignolin in explicit water. Lastly, we develop a procedure to infer the unbiased kinetics from simulations

combining SR and MetaD, showing an improved tradeoff between speedup and accuracy in comparison to MetaD simulations alone.

## Results

### A two wells model

We begin by combining SR with MetaD simulations. A simple model system, where the optimal CV is well defined, is considered first. For this model, combining SR with MetaD leads to greater speedups than MetaD independently, even when biasing with the optimal CV. The speedup is defined as the ratio of the mean number of timesteps before a first-passage is observed, between unbiased and biased simulations. The model is shown in Fig. 1a. It is a two-dimensional harmonic trap, divided into two states centered at $(x = \pm 3, y = 0)$ Å by a barrier of $5 k_B T$ for all y values.

The harmonic trap is soft, such that a particle in one of the wells can easily travel about 50 Å away from the central barrier. The exact parameters of the potential are given in the "Methods" section. They were chosen such that the unbiased mean FPT (MFPT) between the wells is long (7.5 ns), but can still be sampled in unbiased simulations. The optimal CV is the x-coordinate. We follow the trajectories of a particle that was initialized at the right minimum and define the FPT criterion as arriving to the left minimum ($x \leq -3$ Å).

For comparison, we first performed SR on standard MD simulations using different resetting rates. The obtained speedups are shown in Fig. 1b. The speedup increases with the resetting rate, reaches a maximum value of ~4, which is obtained at a rate of $50\,ns^{-1}$, and decreases for higher rates. This non-monotonic trend can be understood since in the limit of high resetting rates all trajectories are restarted before a transition can occur, and the speedup drops to zero.

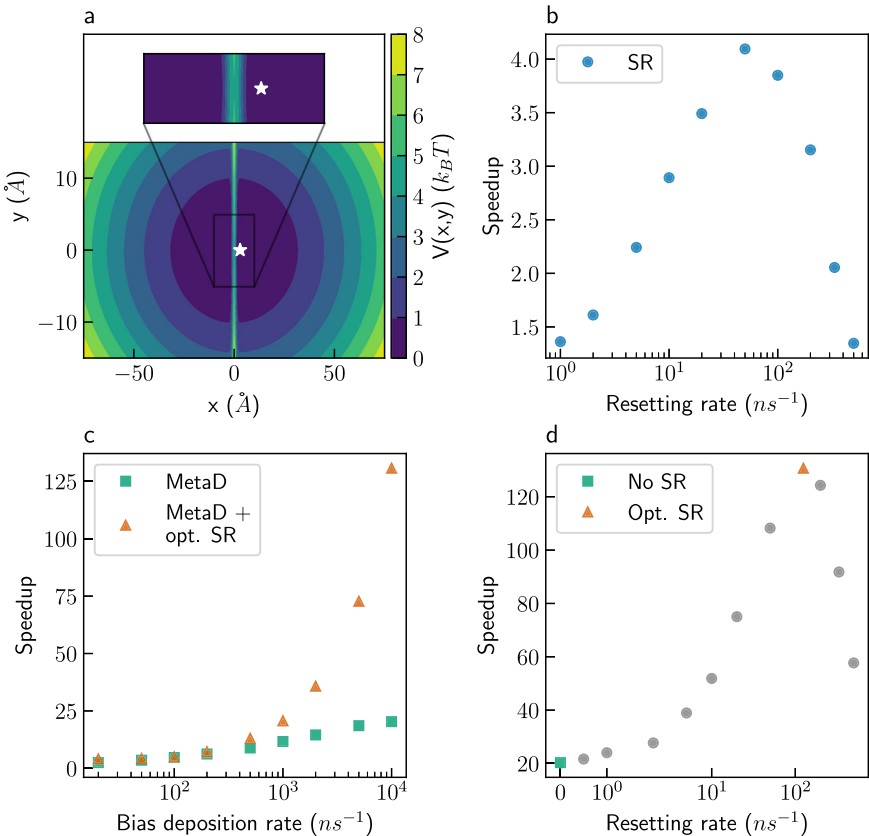

**Fig. 1 | The two wells model. a** The two wells potential. The star marks the initial position. **b** Speedups obtained for stochastic resetting (SR) simulations at different resetting rates. **c** Speedups obtained for Metadynamics (MetaD) simulations at different bias deposition rates (green squares), and for combined MetaD + SR simulations at an optimal (opt.) resetting rate (orange triangles). **d** Speedups obtained for MetaD + SR simulations with a bias deposition rate of $10^4$ ns$^{-1}$ and different resetting rates. Source data are provided as a Source Data file.

The resetting rate which leads to maximum speedup will be referred to as the optimal resetting rate in this paper.

Next, we performed MetaD simulations without SR using the x-coordinate as a CV and varying the bias deposition rates. Other bias parameters are given in the "Methods" section. The results are shown as green squares in Fig. 1c. The speedup increases with the bias deposition rate, with a value of ~20 attained for a bias rate of $10^4$ ns$^{-1}$ (every 100 simulation steps). It is evident that MetaD leads to larger acceleration than SR for this system, giving speedups that are greater by a factor of ~5.

We then combined SR with MetaD and found that even greater speedups are obtained (Fig. 1c, orange triangles). How the combination of resetting and MetaD is done in practice is shown in Fig. 1d, using the highest bias deposition rate ($10^4$ ns$^{-1}$) as an example. The green square in Fig. 1d shows the speedup obtained with MetaD and no resetting. Then, for that given bias deposition rate, we add SR at increasing rates and evaluate the resulting speedup. We stress that the MetaD bias is zeroed at every resetting event. We observe the same qualitative behavior seen in Fig. 1b, with the speedup increasing until some optimal resetting rate, highlighted with an orange triangle. We repeat this procedure for all bias deposition rates, and present the optimal speedup by orange triangles in Fig. 1c. Combining SR with MetaD gave additional acceleration for all bias deposition rates, with a maximal speedup of ~130 at a bias rate of $10^4$ ns$^{-1}$. The corresponding optimal resetting rate was found to be 125 ns$^{-1}$, which is significantly slower than the bias deposition. The fact that SR can further accelerate MetaD simulations, even when performed with optimal CVs, is the first key result of this paper.

Our results show minimal sensitivity to the initial positions (Supplementary Discussion 1). Supplementary Fig. 1 gives results for additional initial positions, showing similar accelerations as in Fig. 1c. We note that we allow high bias deposition rates of up to once every 100 steps, as commonly done in analyzing the speedup in sampling[48]. Much lower rates will be employed when discussing the kinetics inference later. We also acknowledge that very high speedups might come with a cost, e.g., very aggressive non-tempered MetaD can lead to several transitions, but the resulting oscillating bias would make it impossible to converge the desired properties. Resetting actually reduces this risk, by further accelerating the transitions while minimizing the bias deposition.

MetaD practitioners might wonder: (1) How to tell whether SR will accelerate my simulations?, and (2) How to identify the optimal resetting rate and estimate what would be the resulting speedup?

Next, we show that both questions can be answered at almost no additional cost, assuming some MetaD trajectories are already available. To answer the first question, we showed in a recent paper[29] that a sufficient condition for acceleration of MD simulations by SR is that the ratio of the standard deviation to the MFPT (the coefficient of variation, COV) is greater than one[47]. Introducing a small resetting rate is then guaranteed to lead to speedup. We stress that this condition holds also for resetting MetaD simulations, with the added benefit that enhanced sampling generates more transitions, and thus gives a much more reliable estimation of the COV compared to unbiased MD simulations. Moreover, if SR does not accelerate the unbiased simulations significantly, it does not mean that it will not do so for MetaD simulations, since biasing alters the FPT distribution significantly and, consequently, may also change the COV.

Figure 2a shows the COV of MetaD simulations (without resetting) as a function of the bias deposition rate, for the two wells model (Fig. 1a). The COV shows non-monotonic behavior with the bias deposition rate. It starts from a value of 1.05 without resetting, drops to 0.80 at an intermediate bias deposition rate and increases up to a value of 1.44 at the highest biasing rate. This shows that MetaD can increase the COV significantly (allowing for further speedup by resetting).

As for the second question, estimating the optimal resetting rate and the resulting speedup is also straightforward for MetaD simulations. The MFPT under a resetting rate $r$ can be estimated using a simple equation[49],

$$\langle \tau \rangle_r = \frac{1 - \tilde{f}(r)}{r\tilde{f}(r)}. \tag{1}$$

In Equation (1), $\tilde{f}(r)$ is the Laplace transform of the FPT distribution for the MetaD simulations, and $\langle \tau \rangle_r$ is the MFPT for MetaD simulations with SR at resetting rate $r$. The Laplace transform is evaluated as

$$\tilde{f}(r) = \langle e^{-r\tau} \rangle \simeq \frac{1}{N} \sum_{j=1}^{N} e^{-r\tau_j}, \tag{2}$$

where $N$ is the number of MetaD trajectories, and $\tau_j$ is the FPT obtained from trajectory $j$. Figure 2b shows the additional speedups, over MetaD without resetting, estimated using Equation (1) (dotted lines). They are plotted as a function of the resetting rate for the two bias deposition rates highlighted with colored circles in Fig. 2a. It is evident that the estimations match results obtained from simulations (full circles). While the full FPT distribution is required for an exact description of

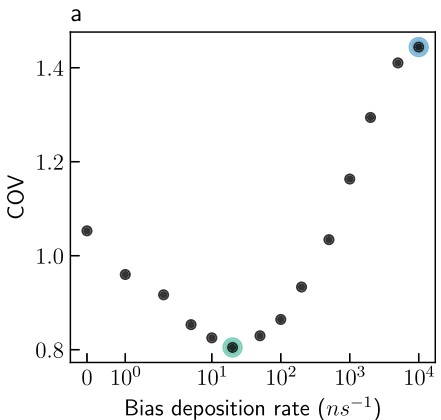

a

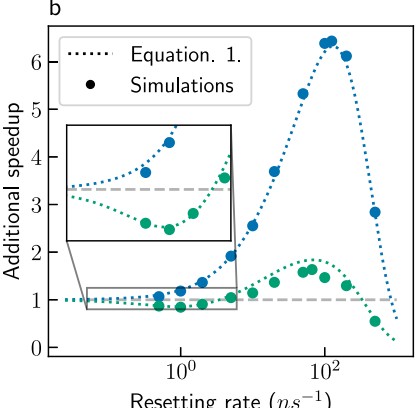

b

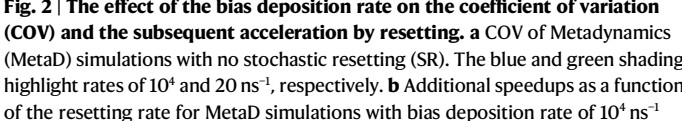

**Fig. 2 | The effect of the bias deposition rate on the coefficient of variation (COV) and the subsequent acceleration by resetting. a** COV of Metadynamics (MetaD) simulations with no stochastic resetting (SR). The blue and green shading highlight rates of $10^4$ and 20 ns$^{-1}$, respectively. **b** Additional speedups as a function of the resetting rate for MetaD simulations with bias deposition rate of $10^4$ ns$^{-1}$ (blue) or 20 ns$^{-1}$ (green), for the two wells model. Full circles present results obtained from simulations, while dotted lines present estimations based on the first-passage time distribution with MetaD and no SR and using Equation (1). The dashed gray line indicates no additional speedup. Source data are provided as a Source Data file.

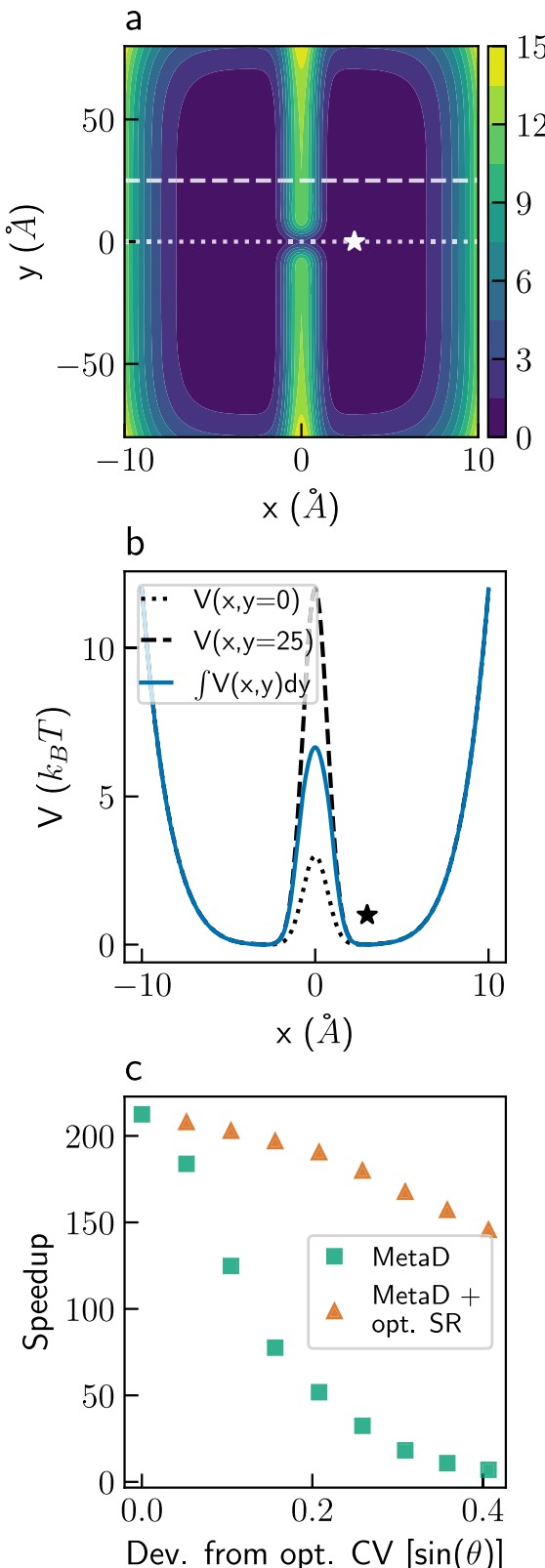

**Fig. 3 | The modified Faradjian-Elber model. a** The modified Faradjian-Elber potential. The star marks the initial position. The dotted and dashed lines mark $y = 0$ and 25 Å, respectively. **b** The integrated projection of the potential on the x-axis (blue), and cross sections of the potential at $y = 0$ and 25 Å (dotted and dashed black lines, respectively). The star marks the initial x-axis value. **c** Speedups obtained for a bias deposition rate of $10^4$ ns$^{-1}$ using suboptimal, rotated collective variables (CVs), in Metadynamics (MetaD) simulations with no stochastic resetting (SR) (green squares) and with optimal (opt.) SR (orange triangles). The angle between the CV and the x-axis serves as a measure for the deviation from the optimal CV and is denoted by $\theta$. Source data are provided as a Source Data file.

introducing a small resetting rate will increase the MFPT. In other words, the COV only indicates the initial slope of the speedup curve as a function of the resetting rate. Interestingly, non-trivial cases where small resetting rates decelerate the process but larger ones accelerate it are also possible, as can be seen for the green curve in the inset of Fig. 2b. For comparison, we also give the results for a bias deposition rate of $10^4$ ns$^{-1}$, for which the COV without SR is > 1, and the initial slope of the speedup with respect to the resetting rate is positive. The results show that, whether the value of the COV is greater or smaller than one, it is worthwhile to estimate the speedup using Equation (1).

To conclude this example, we combined SR with MetaD, leading to greater acceleration than either approach separately. This is demonstrated even for a system for which the optimal CV is known. Since MetaD simulations already enhance the sampling of the underlying process, it is significantly easier to evaluate their COV than for unbiased simulations. If the COV is larger than 1, then MetaD simulations are guaranteed to be further accelerated by SR, and Equation (1) can be used to easily estimate by how much. If not, they may still be accelerated, which can be easily checked using Equation (1).

**The modified Faradjian–Elber potential**

As a second example, we consider a modified version of the two-dimensional potential introduced by Faradjian and Elber when developing the milestoning enhanced sampling method[9]. The potential is shown in Fig. 3a, and full details are given in the "Methods" section. It is also composed of two symmetric wells, with minima at $(x = \pm 3, y = 0)$ Å that are separated by a Gaussian barrier at $x = 0$ Å. The barrier is higher than the first example, $12 k_B T$ for most $y$ values, but has a narrow saddle, only $3 k_B T$ high, around $y = 0$ Å. Figure 3b shows cross sections along the x-axis at $y = 0$ and 25 Å, as well as the effective potential integrated over the entire y-axis.

We follow the trajectories of a particle that was initialized at the right minimum and define the FPT criterion as crossing the barrier and reaching $x < -1$ Å. For this model, we find the same MFPT as in the two wells model. Employing SR on unbiased simulations gave an optimal speedup of ~15 at a resetting rate of 200 ns$^{-1}$. As in the two wells model, MetaD simulations gave higher speedups than SR, with a speedup of ~212 when using the optimal CV, the x-coordinate, at a bias deposition rate of $10^4$ ns$^{-1}$. Using this optimal CV and rate, combining SR with MetaD did not lead to further acceleration of the simulations.

However, in most real systems, the optimal CV is not known, and suboptimal CVs are almost always used[23]. To test the efficiency of SR in such cases, we gradually reduce the quality of the CV by rotating it. The green squares in Fig. 3c show the speedup obtained as a function of the sine of the angle $\theta$ between the CV and the x-axis, which serves as a measure for the deviation from the optimal CV. The degradation in the quality of the CV leads to a decrease in the MetaD speedup with almost no acceleration at an angle of 24°. However, combining SR with MetaD recovers almost all of the speedup of the optimal CV, despite the use of suboptimal CVs. This is shown by the orange triangles in Fig. 3c. Optimizing CVs for condensed phase systems remains a difficult challenge[15,50]. Our results suggest that SR may serve as an alternative, or complementary method, to improving CVs. Instead of using sophisticated algorithms to find better CVs[50–56], one can use SR to

the behavior under SR, we previously showed[29] that as few as a hundred samples are sufficient for estimating the optimal resetting rate.

Surprisingly, for intermediate bias deposition rates, some additional speedup is gained even though the COV without resetting is smaller than 1. This can be seen in Fig. 2b, which shows an optimal speedup of ~2 for a bias deposition rate of 20 ns$^{-1}$, even though its COV without SR is only 0.80. How can it be? A COV < 1 indicates that

obtain a similar speedup at a much lower cost. This is the second key result of this paper.

## Alanine tetrapeptide

Moving on to a molecular system, we demonstrate the capabilities of combining MetaD with SR on alanine tetrapeptide in vacuum. We focus on two of its conformers, "folded" and "unfolded", shown in Fig. 4a. Six dihedral angles serve as important degrees of freedoms, with $\phi_3$ being the slowest one[22,57]. Figure 4b shows the free-energy surface along $\phi_3$ (see the "Methods" section for details), which has two minima separated by an energy barrier of ~15 $k_B T$. Transitions in unbiased simulations from the unfolded state (upper configuration in panel a, left basin in panel b) to the folded one (lower configuration in panel a, right basin in panel b) have an estimated MFPT of ~5.6 μs (Supplementary Discussion 2). Both the folded and unfolded states can each be resolved into four sub-states. An analysis of the effect of combining SR with MetaD on accelerating transitions between these sub-states is provided in Supplementary Discussion 3, with results shown in Supplementary Fig. 2.

To improve the sampling, we performed MetaD simulations using three different CVs: the angle $\phi_3$ serves as the optimal CV, and two adjacent angles, $\phi_2$ and $\psi_3$, serve as suboptimal ones. The two-dimensional free-energy surfaces as a function of all CVs are presented in panels c, d of Fig. 4. They show that $\psi_3$ has some overlap and does

not separate the two states as well as $\phi_3$, while there is almost no separation of the states in $\phi_2$. The simulations were initialized from a fixed, unfolded configuration, marked with stars in panels b-d. The first-passage criterion ($0.5 < \phi_3 < 1.5$ rad) is also marked in these panels, with vertical dashed lines.

Figure 4e shows the speedup of MetaD simulations using different protocols, without SR (green) and with optimal SR (orange). COV values for simulations with no SR are given in panel f. As expected, using $\phi_3$ as a CV gives the greatest speedup, and a COV of ~0.3 for which the optimal resetting rate is $r^* = 0$. Thus, there is no benefit from SR in this case. Suboptimal CVs show similar behavior to that observed for the Faradjian-Elber potential, but for a realistic system: The speedups obtained for MetaD without SR decrease when using bad CVs and the COV values increase above one. Namely, while MetaD simulations using $\phi_3$ as the CV gives more than four orders of magnitude speedup, simulations using $\psi_3$ and $\phi_2$ lead to accelerations by factors of only ~580 and ~4, respectively. Concurrently, the COV of $\psi_3$ is ~1.24 while for the worst CV, $\phi_2$, it is ~3. SR becomes more effective the poorer the CV is, giving an additional speedup of ~133 over MetaD when using $\phi_2$ as a CV.

## Chignolin in water

As a final example, we demonstrate our method on a more complicated system, the mini-protein chignolin in explicit water (5889 atoms

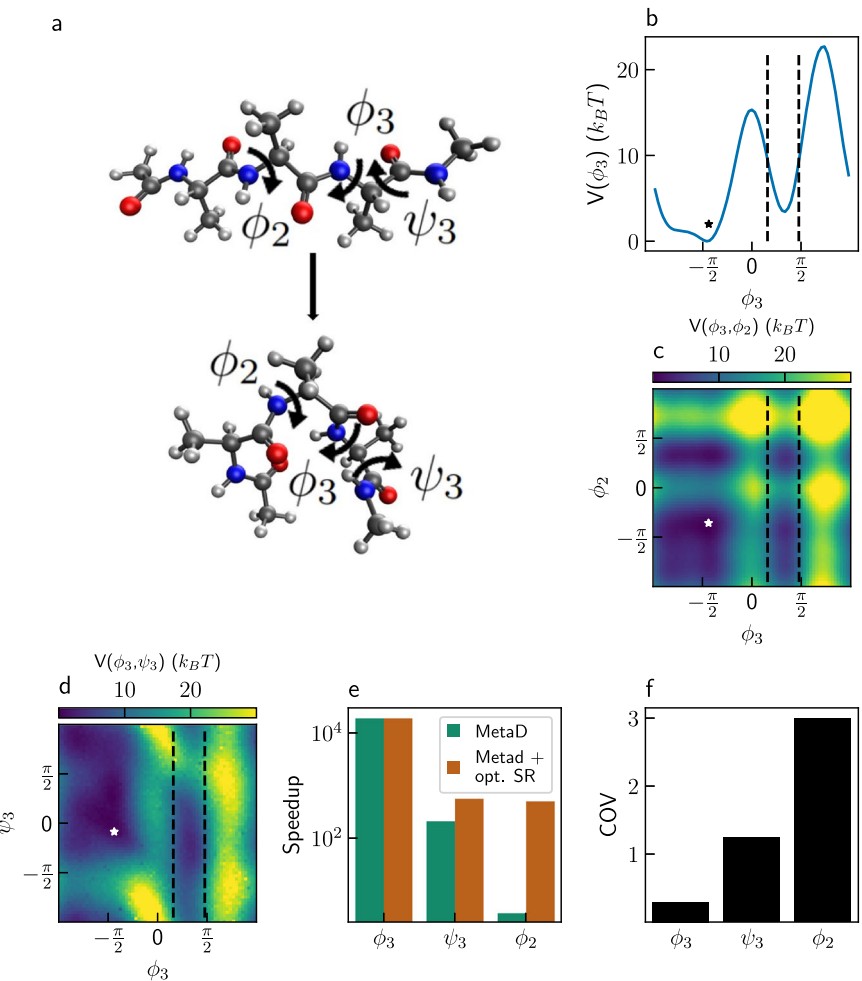

**Fig. 4 | Alanine tetrapeptide. a** Two conformers of alanine tetrapeptide. The white, gray, blue, and red balls represent hydrogen, carbon, nitrogen, and oxygen atoms, respectively. **b** One-dimensional free-energy surface of alanine tetrapeptide along $\phi_3$. **c** Two-dimensional free-energy surface of alanine tetrapeptide along $\phi_3$ and $\phi_2$. **d** Two-dimensional free-energy surface of alanine tetrapeptide along $\phi_3$ and $\psi_3$. The stars in panels b-d mark the initial configuration. **e** Speedup of MetaD simulations, without stochastic resetting (SR) (green) and with optimal (opt.) SR (orange). **f** Coefficient of variation (COV) of MetaD simulations without SR for different collective variables. Source data are provided as a Source Data file.

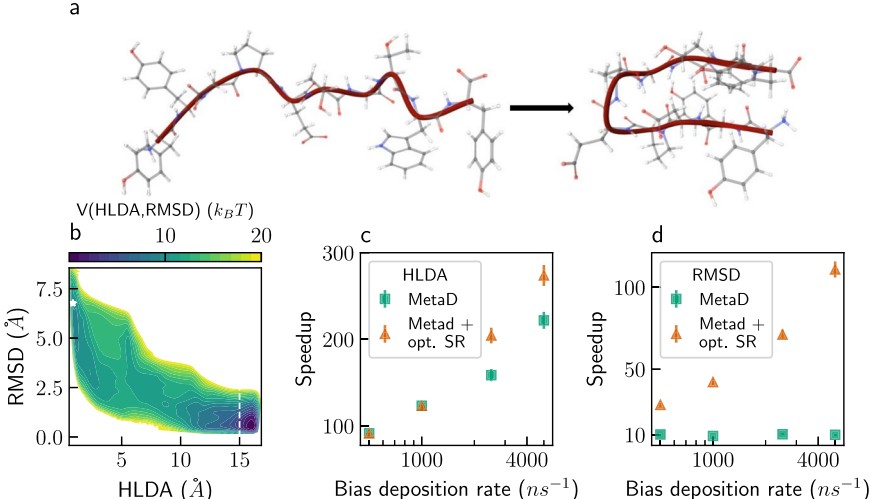

**Fig. 5 | Chignolin in water. a** Ball-and-stick representation of the unfolded and folded states of chignolin in explicit water solvent (5889 atoms), including a cartoon representation of the backbone in crimson. The white, gray, blue, and red spheres represent hydrogen, carbon, nitrogen, and oxygen atoms, respectively. **b** Two-dimensional free-energy surface of chignolin (see the "Methods" section for details), along a collective variable (CV) based on harmonic linear discriminant analysis (HLDA), and the C-alpha root-mean-square deviation (RMSD) from a folded configuration. The white star marks the initial configuration and the dashed line indicates the first-passage criterion. The white regions have free energy > 20 $k_BT$. **c** Speedup as a function of bias deposition rate for Metadynamics (MetaD) simulations using the HLDA-based CV, without stochastic resetting (SR) (green) and with optimal (opt.) SR (orange). **d** Speedup as a function of bias deposition rate for MetaD simulations using the RMSD as CV, with and without SR. Error bars represent standard deviation divided by the square-root of the number of independent trajectories ($n = 1000$). Source data are provided as a Source Data file.

in total), following transitions from an unfolded to a folded state (Fig. 5a). A linear combination of six interatomic contacts, optimized via harmonic linear discriminant analysis (HLDA) by Mendels et al.[56], serves as a good CV. The C-alpha root-mean-square deviation (RMSD) from a folded configuration serves as a suboptimal CV. Figure 5b shows the two-dimensional free-energy surface along the chosen CVs, with the white star representing the initial configuration (shown on the left of Fig. 5a), and the dashed line marking the first-passage criterion. See the "Methods" section for the full details of how the free energy was constructed. The unbiased mean folding time is ~305 ns.

We performed MetaD simulations with bias deposition rates of $5 \times 10^2$–$5 \times 10^3$ $ns^{-1}$, corresponding to every 100–1000 timesteps. Using the HLDA-based CV, we obtained speedups of two orders of magnitude, as shown in Fig. 5c (green squares). Combining with SR further accelerates simulations with high bias deposition rates, by ~25%, using optimal resetting rates (orange triangles). Using the C-alpha RMSD as CV, MetaD gives speedups of only one order of magnitude, as shown in Fig. 5d (green squares). This is because the RMSD-based CV does not separate the states as well as the HLDA-based CV, as can be seen in the free-energy surface. Yet, combining SR with MetaD simulations using this suboptimal CV leads to an additional speedup of up to ~11.2 over the MetaD simulations. This shows that combining SR with MetaD for a suboptimal CV leads to similar acceleration as for the good CV, but at slightly higher bias deposition rates.

**Kinetics inference**

To conclude the paper, we demonstrate that SR can improve the inference of unbiased kinetics from MetaD simulations, using the case of alanine tetrapeptide as an example. We emphasize that the inference of free-energy surfaces using SR and MetaD is an open problem, that is beyond the scope of this work.

The unbiased FPT distribution can be extracted from MetaD simulations with no SR through a procedure known as infrequent MetaD (iMetaD)[2]. In this method, the MFPT is obtained by rescaling the FPT of each MetaD trajectory by an acceleration factor that depends exponentially on the external bias (see Equation S3 in Supplementary Discussion 4). iMetaD assumes that no bias is deposited near the

transition state, and that none of the basins are over-filled. When this assumption is valid, the distribution of the rescaled FPTs matches the unbiased distribution. However, the assumption does not hold for suboptimal CVs or high bias deposition rates[1], which result in over-deposition. Due to the exponential dependence of the acceleration factor on the bias, trajectories exhibiting over-deposition result in very large acceleration factors. They contribute unrealistically long FPTs to the rescaled distribution, shifting the obtained MFPT from the true value by orders of magnitude.

The inference can be improved, even with suboptimal CVs, by decreasing the bias deposition rate, resulting in a tradeoff between speedup and accuracy. This tradeoff is demonstrated in Fig. 6. It shows (green squares) the error in the estimation of the unbiased MFPT as a function of speedup, for iMetaD simulations of alanine tetrapeptide biasing the $\psi_3$ angle, which is a suboptimal CV. The prediction error is defined as $|\langle\tau\rangle_{true} - \langle\tau\rangle_{est}|/\langle\tau\rangle_{true}$ where $\langle\tau\rangle_{true}$ is the true unbiased MFPT and $\langle\tau\rangle_{est}$ is the estimated MFPT.

Next, we demonstrate that resetting can give a better tradeoff, reducing the error for all speedups, as shown by the orange triangles in Fig. 6. These results were obtained by adding SR at different resetting rates to the iMetaD simulations at the highest bias deposition rate (highlighted with a gray circle).

Full details explaining how to infer the unbiased MFPT from combined SR and iMetaD simulations are given in Supplementary Discussion 4. Here, we briefly provide only the key ingredients and underlying intuition. We note that between resetting events, the trajectories are standard iMetaD simulations. Moreover, due to SR, the short trajectories between restarts, can be treated as independent from one another (recall that restart also zeros previous bias). As a result, we can use the standard iMetaD rescaling procedure on each short trajectory, and then evaluate the unbiased survival probability at short times. For short enough times, even with suboptimal CVs, we will avoid over-deposition and get a good estimate of the unbiased survival. Finally, we assume that the survival probability decays exponentially, as commonly done in iMetaD[1], and obtain an estimate of the unbiased MFPT from its slope. The quality of the linear fit can be used to assess the reliability of the predicted MFPT, similar to the

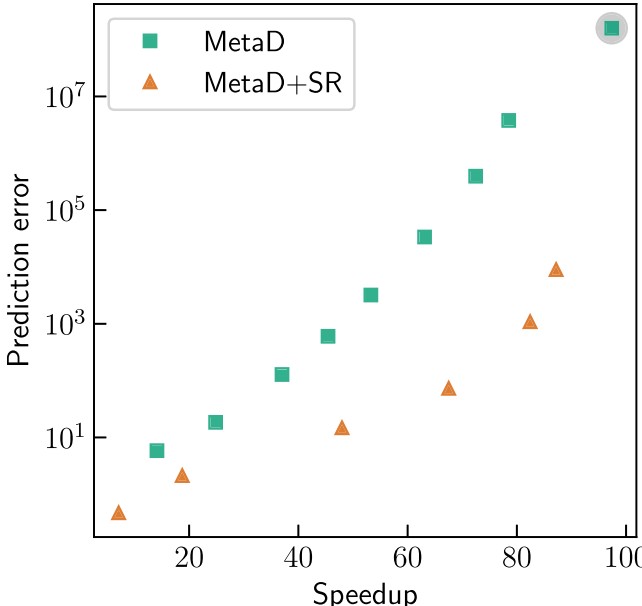

**Fig. 6 | Kinetics inference.** The error in the prediction of the unbiased mean first-passage time of alanine tetrapeptide as a function of speedup for infrequent Metadynamics (iMetaD) simulations at different bias deposition rates, biasing the $\psi_3$ angle (green squares). The results for a bias deposition rate of 50 ns$^{-1}$ are highlighted with a gray circle. For this rate, we also performed iMetaD simulations with stochastic resetting (SR) at different resetting rates (orange triangles). Source data are provided as a Source Data file.

Kolmogorov–Smirnov test suggested by Salvalaglio et al. in standard iMetaD simulations[1,58,59].

Our results show that, for alanine tetrapeptide, applying SR to iMetaD simulations and using the proposed inference procedure gives a better tradeoff than decreasing the bias deposition rate. In choosing between the two, practitioners of iMetaD might consider a simple question: given a fixed simulation time, would the error obtained be lower in a single long iMetaD trajectory or on a series of shorter iMetaD trajectories with SR? For suboptimal CVs, the acceleration factor becomes increasingly unreliable with time, due to over-deposition of the external bias. In this case, short trajectories with minimal bias are preferred, and SR may be more favorable.

## Discussion

In this work, we combine SR with MetaD simulations. We show that resetting can further accelerate MetaD simulations, even when the optimal CV is used. In practice, the optimal CV is almost never known and suboptimal CVs are employed. We provide examples in which adding SR to MetaD simulations performed with poor CVs, leads to speedups comparable to using the optimal CV. This suggests that resetting may serve as an alternative to the challenging task of improving suboptimal CVs using sophisticated algorithms.

Resetting can be easily implemented in existing MD codes, and is highly parallelizable, as the trajectories between resetting events are entirely independent. Furthermore, the implementation of resetting is agnostic to the details of the simulated system. It is the same for any kind of first-passage process, though it may not be beneficial in all cases. Fortunately, testing whether SR can accelerate simulations is very easy. Given a small number (-100) of short MetaD trajectories, showing one transition each, we can estimate the COV and find whether SR would further accelerate the simulations, and by how much, using Equation (1). Resetting can be of benefit even when it does not provide additional acceleration on top of the one attained by MetaD. We demonstrate that SR can improve the inference of the unbiased kinetics from iMetaD simulations performed with suboptimal CVs,

giving a better tradeoff between speedup and accuracy for alanine tetrapeptide.

Finally, we conjecture that benefits coming from combining MetaD and SR are not limited to the examples presented herein, and are much more general. The reason is that MetaD, and similar methods, flatten the free-energy surface. Previous work has shown that SR is particularly efficient for flat landscapes[40,60,61], with the extreme case being free diffusion[62]. Thus, starting from an arbitrary free-energy surface, MetaD changes it to one that is more amenable to acceleration by SR. Future method development would likely harness the power of this important observation.

## Methods

### Model potentials

Here, we present the exact equations and parameters of the chosen model potentials. The parameters are given such that spatial distances are in Å and potential energies are in units of $1\,k_B T$ for a temperature of 300 $K$.

The two wells model is described by Equation (3), with $A_1 = 1 \times 10^{-3}$, $A_2 = 1 \times 10^{-2}$, $B = 5$, $C = 1$.

$$V(x,y) = A_1 x^2 + A_2 y^2 + B \exp\left(-Cx^2\right) \tag{3}$$

For the modified Faradjian-Elber potential, we used Equation (4), with $A_1 = 1.2 \times 10^{-5}$, $A_2 = 12$, $B = 0.75$, $\sigma_1 = 1$, $\sigma_2 = 0.5$, and $y' = 0.1y$.

$$V(x,y) = A_1\left(x^6 + y'^6\right) + A_2 \exp\left(-\frac{x^2}{\sigma_1^2}\right)\left[1 - B \exp\left(-\frac{y^2}{\sigma_2^2}\right)\right] \tag{4}$$

The modifications were made to: (1) stretch the y-axis, and (2) ensure there is a barrier also at $y = 0$ Å by setting the value of $B \neq 1$.

### General simulations details

Initial velocities were sampled from the Maxwell–Boltzmann distribution, while initial positions were fixed (equivalent to sampling from a delta function positions distribution). We defined the FPT as the earliest instance at which a certain criterion was met, as specified above for each system. The COMMITTOR command in PLUMED was used for testing this criterion and stopping the simulations when it was fulfilled. For most simulations with SR, we sampled the time intervals between resetting events from an exponential distribution with a fixed resetting rate (Poisson resetting) using Python. Simulations designated for kinetics inference used constant time intervals between resetting events (sharp resetting). If a first-passage event did not occur prior to the next resetting time, the simulation was restarted. We stress that we continue tallying the overall time until a first-passage occurred, regardless of the number of resetting events in between. For simulations combining SR with MetaD, we emphasize that the MetaD bias potential was zeroed after each resetting event.

We sampled $10^4$ independent trajectories in all sets of simulations, except for simulations of chignolin, where we sampled $10^3$ trajectories. In all plots, the standard error was calculated as the standard deviation of the observable, divided by square-root of the number of independent trajectories. In all cases, it was found to be smaller than the symbol size, except for Fig. 5, where the error is presented using vertical bars.

### Simulations details for model systems

Simulations of model potentials were performed in the Large-scale Atomic/Molecular Massively Parallel Simulator (LAMMPS)[63]. All of them were performed in the canonical (NVT) ensemble at a temperature $T = 300$ K, using a Langevin thermostat with a friction coefficient $\gamma = 0.01$ fs$^{-1}$. The integration time step was 1 fs. We followed the trajectories of a single particle with mass $m = 40$ g mol$^{-1}$, representing an argon atom.

MetaD was implemented using PLUMED 2.7.1[64–66]. We used a bias factor of 10, bias height of $0.5k_BT$ and grid spacing of 0.01 Å. The Gaussians width was $\sigma = 1.3, 0.15$ Å for the two wells model and the modified Faradjian-Elber potential, respectively.

## Simulations details for molecular systems

For the simulations of alanine tetrapeptide, we used input files by Invernizzi and Parrinello[22], given in PLUMED-NEST, the public repository of the PLUMED consortium[65], as plumID:22.003. For the simulations of chignolin, we used input files by Ray et al.[28], also given in PLUMED-NEST, as plumID:22.031. All simulations were performed in GROMACS 2019.6[67] patched with PLUMED 2.7.1[64–66]. They were carried out in the NVT ensemble at temperatures of 300 K and 340 K for alanine tetrapeptide and chignolin, respectively, using a stochastic velocity rescaling thermostat[68], and integration time step of 2 fs. Chignolin was solvated in 1907 water molecules, and two sodium atoms were added to neutralize the system. Additional setup details can be found in the appropriate PLUMED-NEST repositories.

We used a bias height of $0.5\,k_BT$ and grid spacing of 0.001 rad for all MetaD simulations of alanine tetrapeptide. The bias width $\sigma$ was taken as 10% of the unbiased fluctuations within the narrowest wells, 0.013, 0.013, 0.05 rad for angles $\phi_3$, $\phi_2$, and $\psi_3$, respectively. In most simulations, we used a bias factor of 20 and bias deposition rate of $5 \times 10^3\,\text{ns}^{-1}$, updating the bias potential every 100 timesteps. When performing iMetaD, we used a smaller bias factor of 10. This bias factor was also used for the results presented in Supplementary Discussion 3. In MetaD simulations of chignolin, we used a bias height of $0.5\,k_BT$, bias factor of 15, $\sigma = 0.5$ Å, and grid spacing of 0.05 Å, for both CVs.

## Free-energy surfaces

Free-energy surfaces (FES) of alanine tetrapeptide were obtained through reweighted histograms of MetaD simulations[14,15]. The two-dimensional FES along the $\phi_3$ and $\phi_2$ dihedral angles (Fig. 4c of the main text) uses a 500 ns long simulation, biasing both angles. The bias height was $0.5\,k_BT$, the bias width was 0.35 rad for both angles and the bias factor was 20. Bias deposition rate was $5 \times 10^3\,\text{ns}^{-1}$, updating the bias every 100 timesteps. This simulation is also used for the one-dimensional FES along $\phi_3$ in Fig. 4b, and for the FES of Supplementary Fig. 2a. The two-dimensional FES of Fig. 4d uses a 50 ns long simulation biasing the $\phi_3$ and $\psi_3$ angles. All MetaD parameters were as specified above, except the bias width, which was 0.05 rad for both CVs. All other simulation details are as specified in the previous section.

The FES of chignolin was obtained using umbrella sampling[4,5], constraining both the CV based on harmonic linear discriminant analysis (HLDA) and the CV based on C-alpha root-mean-square deviation (RMSD) from a folded configuration. We performed 256, 100 ns long simulations with harmonic constraints centered at all combinations of $\text{HLDA} \in \{1, 2, \ldots 16\}$ Å and $\text{RMSD} \in \{0.5, 1, \ldots 8\}$ Å. We used harmonic constant of $k = 3\,k_BT\,\text{Å}^{-2}$ for both CVs. The FES was constructed through the Weighted Histogram Analysis Method (WHAM) algorithm, using the implementation of Grossfield[69].

## Reporting summary

Further information on research design is available in the Nature Portfolio Reporting Summary linked to this article.

## Data availability

Source data are provided with this paper. All data was also deposited on GitHub under the https://doi.org/10.5281/zenodo.10210352[70]. Source data are provided with this paper.

## Code availability

Input files for simulations, and example scripts for the implementation of resetting in Python, are available in the GitHub repository[70].

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

## Acknowledgements

We acknowledge support by the Israel Science Foundation (grants No. 1037/22 and 1312/22 to B.H. and grant No. 394/19 to S.R.) and the Pazy Foundation of the IAEC-UPBC (grant No. 415-2023 to B.H.). This project has received funding from the European Research Council (ERC) under the European Union's Horizon 2020 research and innovation program (grant agreement No. 947731 to S.R.).

## Author contributions

O.B. performed all simulations and data analysis. S.R. and B.H. designed and supervised the research. O.B. S.R. and B.H. wrote the manuscript.

## Competing interests

The authors declare no competing interests.
