## [Peer Review File · Nature Communications]

Combining stochastic resetting with Metadynamics to speed-up molecular dynamics simulationsReviewer #2 (Remarks to the Author):

This manuscript presents an articulated illustration of a MD techniques, aimed at accelerating processes in which slow modes hamper sampling efficiency. By fine-tuning their method based on stochastic resetting (SR MD), the authors demonstrate on the one hand very good to excellent performance (measured in terms of MD acceleration) of the method applied in a stand-alone way, but also in combination with metadynamics, which is biasing appropriate collective variables that may be hard to be defined a priori. Being able to work "CV-free", SR MD can therefore accelerate metaD, especially in a regime of suboptimal CV choice. Broadly, this offers additional means to combine SR MD with other flavours of accelerated MD. The authors provide the tools for choosing the only relevant parameter of their method, the resetting frequency, delivering this way a roadmap to promote this method to a standard, powerful tool for computation around rare events, including the extraction of unbiased kinetic parameters.

While, based on the above, I'm supportive of this method becoming more and more popular, not least by the acceptance of this manuscript for publication in this journal, I have a few questions:

a) The examples chosen serve the purpose of methodological benchmarking, and while they admit large activation barriers, they are poorly representative of real life problems, I'm thinking for example of solid-solid phase transitions, which incidentally are mentioned by the authors. What would be the implications, in terms of efficiency/inefficiency for this class of rare-event problems? Can the given protocol be transferable to that class of problems?

b) In example #2, SR does not manage to overtake MetaD, why is it so ? Does the optimal CV here behave differently from the optimal CV in example #1 ?

c) Why does SR work in MetaD, where the bias is zeroth at every resetting step ? I can think of the potential energy of any MetaD-biased chain to increase on the average, by design, so resetting the bias would corresponds to "letting go" of a configuration at higher potential energy with a likley high rate of conversion into kinetic energy, which will benefit the SR approach by making barrier crossing more likely ? To what extent SR will increase fluctuations along CV directions then ?

d) Is there any role of the thermostat in this approach ? Is the thermostat coupling to the system part of the reasoning that will need to enter the choice of the resetting protocol ?

e) Why is SR working in general ? The condition of a wide (enough) distribution of transition times is clear, however can one reason in terms of system response? If we think of resetting as a perturbation, will the system stay in a non-equilibrium state then (between resettings)? Can the resetting frequency be related to system response time ?

I found the SR method illustration rather clear, and the SI useful to reproduce and further expand the method.

Reviewer #3 (Remarks to the Author):

The authors show how the recently proposed stochastic resetting (SR) method can be combined with the popular enhanced sampling method metadynamics (MetaD), to give rise to a novel and efficient enhanced sampling scheme for molecular simulations.

Given the paramount importance of the sampling problem to molecular simulations, any advance in this regard is extremely valuable to the community.

Stochastic resetting takes advantage of the fact that in many complex systems, the mean time required for an interesting transition to happen is larger than the median time. It is intuitive that, under these conditions, running several short MD trajectories will allow more transitions to be observed than running longer ones. The authors show that the rigorous formulation of SR allows to

go beyond this simple argument, and even provide improvements to the well-established MetaD method. Of particular relevance is the fact that SR can bring significant improvements when it comes to estimating kinetic rates.

The paper is well written, and clearly presents the proposed method supporting it with several examples. It introduces a fresh perspective on the much-studied problem of enhanced sampling, which has the potential to advance the field and become relevant to the broad community of MD practitioners. It clearly deserves publication.

The following are some minor points that if addressed could improve the present manuscript:

1. For completeness, the introduction should cite enhanced sampling methods that also rely on restarting trajectories, such as milestoning and weighted ensembles.
2. A brief discussion should be devoted to how the choice of the starting point influences the COV and thus the speedup. Even better, a figure could be added for one of the 2D models, with a heatmap of the COV as a function of the starting coordinates.
3. The term "speedup" can be ambiguous. It should be specified in the main text and/or in the figure caption that it refers to the frequency of transitions observed.
4. A discussion analogous to the following one should be added, if possible. This "speedup" does not in general correlate with the computational cost of estimating observables such as the free energy difference between two states. For example, a very aggressive non-tempered MetaD can lead to several transitions, but the resulting oscillating bias would make it impossible to estimate the free energy. Unless a good CV is used, such MetaD run would also not sample the correct transition state [16]. On the other hand, SR not only brings to more transitions per simulated time step, but it can in principle be used to estimate free energies. Furthermore, it does not alter the transition state, and in fact even allows to compute kinetic rates, as shown.
5. More emphasis could be placed on the fact that the proposed SR method is embarrassingly parallelizable, which is an extremely important characteristic.
6. There is a typo in the SI, "longer then" -> "longer than"

REVIEWER COMMENTS

Reviewer 1:

The work by Hirshberg and collaborators is very interesting, striking and possible to achieve great impact. I do have certain concerns though (one major, one minor) that should be addressed before I think this work is ready for Nature Comm

Major: fig 3c is interesting in showing how biasing even along y still works with the new approach. I think this is because there are only 2 degrees of freedom. So the resetting with randomized starts finds the right escape. The more important result is that for the tetra peptide. Here there are many degrees of freedom and the authors provide some results which are in right direction but I am not convinced. I hope the authors can analyze their data without having to run extra simulations and convince me. The tetra peptide has at least 8 states of interest. The increase/decrease in MFPTs relative to unbiased MD and ordinary metadynamics need to be compared for multiple state to state transitions not just ϕ_3 - see for eg fig 4 in J. Chem. Theory Comput. 2021, 17, 11, 6757–6765 <https://pubs.acs.org/doi/full/10.1021/acs.jctc.1c00431> where some ~ 20 transitions were compared. Analysis similar to this should be carried out also for free energy convergence along the other dihedrals. This becomes extremely important in light of lack of a rigorous proof for convergence of SR+metadynamics

Minor: In figure 1, a bias deposition rate of $1/\text{ps}$ or $10^3/\text{ns}$ is already on the upper end of aggressive biasing in metadynamics. Thus the x-axis of fig 1c is in physically not very relevant regimes - and shows speedups which will never be quite practical. It should be reduced to max of $2 \times 10^3/\text{ns}$. Similar plots should be provided for the other systems especially the tetra peptide.

In conclusion I am excited by the method but believe the authors should do a better job of convincing in the high dimensional case. Once that is done I would be happy to recommend publication

- Pratyush Tiwary, University of Maryland

Reviewer #1

The work by Hirshberg and collaborators is very interesting, striking and possible to achieve great impact. I do have certain concerns though (one major, one minor) that should be addressed before I think this work is ready for Nature Comm.

Response: Thank you for your very positive evaluation of our manuscript and for recommending it for publication in Nature Communications subject to revision. We thoroughly addressed your concerns below, and hope that you will now find our work ready for publication.

Major: fig 3c is interesting in showing how biasing even along y still works with the new approach. I think this is because there are only 2 degrees of freedom. So the resetting with randomized starts finds the right escape. The more important result is that for the tetra peptide. Here there are many degrees of freedom and the authors provide some results which are in right direction but I am not convinced. I hope the authors can analyze their data without having to run extra simulations and convince me. The tetrapeptide has at least 8 states of interest. The increase/decrease in MFPTs relative to unbiased MD and ordinary metadynamics need to be compared for multiple state to state transitions not just ϕ_3 - see for eg fig 4 in J. Chem. Theory Comput. 2021, 17, 11, 6757–6765 <https://pubs.acs.org/doi/full/10.1021/acs.jctc.1c00431> where some ~20 transitions were compared.

Response: In the original manuscript, we defined the first passage criterion using the ϕ_3 angle only. However, resetting is very general: we can define the first passage in terms of other CVs, and apply resetting in the same manner.

Following the reviewer's request, we added to the SI (see section 6 and Figures S2 of the revised SI) an analysis of additional transitions between the most stable state to three other low-energy states. We tested different CVs and bias deposition rates. We find that combining SR with MetaD expedites most of the cases, especially when suboptimal CVs are used, but resetting is not guaranteed to help in every scenario. We also added a comment in the main text referring to this new analysis on p. 5 of the revised manuscript: "Each of the states can be resolved to 4 sub-states. An analysis of the effect of combining SR with MetaD on accelerating transitions between those sub-states is provided in the supporting information (Section 6)."

Analysis similar to this should be carried out also for free energy convergence along the other dihedrals. This becomes extremely important in light of lack of a rigorous proof for convergence of SR+metadynamics.

Response: In principle, SR can also be used to obtain free energy surfaces, but the required inference procedure is an open problem in the field. We intend to pursue this issue in the near future. To avoid further confusion, we emphasize this important issue on

page 8 of the revised manuscript: “We emphasize that the inference of free-energy surfaces using SR and MetaD is an open problem, that is beyond the scope of this work”.

Minor: In figure 1, a bias deposition rate of 1/ps or $10^3/\text{ns}$ is already on the upper end of aggressive biasing in metadynamics. Thus the x-axis of fig 1c is in physically not very relevant regimes - and shows speedups which will never be quite practical. It should be reduced to max of $2 \times 10^3/\text{ns}$.

Response: The reviewer is correct, this rate is very aggressive for kinetics inference. But in this plot, we are analyzing the speedup in sampling first, for which we use a maximal bias deposition rate of once every 100 steps, which is commonly used (and advised in tutorials such as https://www.plumed.org/doc-v2.8/user-doc/html/m_e_t_a_d.html and <http://cgmartini.nl/index.php/tutorials-general-introduction-gmx5/metadynamics>).

We emphasize this important point in the revised manuscript on page 3 in a comment that reads: “We note that we allow high bias deposition rates of up to once every 100 steps, as commonly done in analyzing the speedup in sampling [48]. Much lower rates will be employed when discussing kinetics inference below.”

Similar plots should be provided for the other systems especially the tetra peptide.

Response: Plots for the tetrapeptide were added to the SI.

In conclusion I am excited by the method but believe the authors should do a better job of convincing in the high dimensional case. Once that is done I would be happy to recommend publication.

Response: After addressing all of the reviewer’s concerns we hope that you will now find our paper suitable for publication.

Reviewer #2

This manuscript presents an articulated illustration of a MD techniques, aimed at accelerating processes in which slow modes hamper sampling efficiency. By fine-tuning their method based on stochastic resetting (SR MD), the authors demonstrate on the one hand very good to excellent performance (measured in terms of MD acceleration) of the method applied in a stand-alone way, but also in combination with metadynamics, which is biasing appropriate collective variables that may be hard to be defined a priori. Being able to work "CV-free", SR MD can therefore accelerate metaD, especially in a regime of suboptimal CV choice. Broadly, this offers additional means to combine SR MD with other flavours of accelerated MD. The authors provide the tools for choosing the only relevant parameter of their method, the resetting frequency, delivering this way a roadmap to promote this method to a standard, powerful tool for computation around rare events, including the extraction of unbiased kinetic parameters. While, based on the above, I'm supportive

of this method becoming more and more popular, not least by the acceptance of this manuscript for publication in this journal.

Response: Thank you for the very positive review of our manuscript, and for finding it suitable for publication in Nature Communications.

I have a few questions:

a) The examples chosen serve the purpose of methodological benchmarking, and while they admit large activation barriers, they are poorly representative of real life problems, I'm thinking for example of solid-solid phase transitions, which incidentally are mentioned by the authors. What would be the implications, in terms of efficiency/inefficiency for this class of rare-event problems? Can the given protocol be transferable to that class of problems?

Response: Nothing is limiting SR to conformational changes. The same procedure could also be applied to solid-solid phase transitions or any other first-passage processes. The only question is whether further acceleration can be obtained or not, but this can be easily checked by evaluating the coefficient of variation (COV) of the first passage time distribution of the unbiased process. If it is larger than unity, SR is guaranteed to accelerate, see ref. [47]. In addition, the implementation in PLUMED is general and can be used with any CV. Therefore, we anticipate that the community will pick this method up in various applications. While we are not sure that solid-solid phase transitions will be accelerated, we anticipate that crystallization from the melt will be. The reason is that solid-solid transitions are between two narrow basins, and therefore the COV criterion might not be satisfied, whereas crystallization processes have characteristic broad basins in the melted state and narrow basins in the crystal. A discussion regarding this issue is added to page 10, which reads: "the implementation of resetting is agnostic to the details of the simulated system. It is the same for any kind of first-passage process, though it may not be beneficial in all cases."

That being said, we present new results in the revised paper, that show a much larger system (the mini protein chignolin) and a different physics of simulations in explicit solvent. This provides clear evidence of the broad applicability of our approach.

b) In example #2, SR does not manage to overtake MetaD, why is it so ? Does the optimal CV here behave differently from the optimal CV in example #1 ?

Response: Yes, SR may show very different behavior for different CVs, even if both of them are optimal. The key factor is the value of the COV for transitions along the CV. Generally, we expect the COV condition to be satisfied for broad basins. In the second example, the basin is much narrower along the optimal CV compared to the first example, which can explain why it is not accelerated by SR.

c) Why does SR work in MetaD, where the bias is zeroth at every resetting step ? I can think of the potential energy of any MetaD-biased chain to increase on the average, by design, so resetting the bias would corresponds to "letting go" of a configuration at higher potential energy with a likely high rate of conversion into kinetic energy, which will benefit the SR approach by making barrier crossing more likely ? To what extent SR will increase fluctuations along CV directions then ?

Response: Thank you for pointing out this highly non-trivial issue. As mentioned in page 2, between resetting events the bias is accumulated normally. This has two consequences: 1) It accelerates the transition thereby reducing the mean first passage time. 2) it also significantly flattens the free energy surface. It was shown that SR is particularly efficient for flat landscapes, see refs. [40, 60, 61]. Therefore, SR is exactly suited to complement the speedup obtained by MetaD and provide further acceleration. A comment on this issue appears in the revised manuscript, on page 10: "Previous work has shown that SR is particularly efficient for flat landscapes, with the extreme case being free diffusion. Thus, starting from an arbitrary free energy surface, MetaD changes it to one that is more amenable to acceleration by SR."

d) Is there any role of the thermostat in this approach ? Is the thermostat coupling to the system part of the reasoning that will need to enter the choice of the resetting protocol ?

Response: Between resetting events, the simulations are standard MetaD simulations. Therefore, the role of the thermostat in our method is the same as in standard MetaD.

e) Why is SR working in general ? The condition of a wide (enough) distribution of transition times is clear, however can one reason in terms of system response? If we think of resetting as a perturbation, will the system stay in a non-equilibrium state then (between resettings)? Can the resetting frequency be related to system response time ?

Response: The question of why resetting works, in general, can be traced back to the inspection paradox from probability theory. We refer the reviewer to ref. [47], where the inspection paradox and its origins are reviewed, and the insight gained is used to explain why, and under which conditions, stochastic resetting expedites the completion of random processes. Importantly, this is done with elementary mathematical tools which help develop a probabilistic intuition for stochastic resetting and how it works.

It is also true that stochastic resetting leads to the emergence of a non-equilibrium steady state, and we refer to a review on the subject (ref. [30]). Yet, in the context of resetting, we are unaware of studies that linked first-passage properties with steady-state properties.

I found the SR method illustration rather clear, and the SI useful to reproduce and further expand the method.

Response: Thank you again for your very positive review.

Reviewer #3 (Remarks to the Author):

The authors show how the recently proposed stochastic resetting (SR) method can be combined with the popular enhanced sampling method metadynamics (MetaD), to give rise to a novel and efficient enhanced sampling scheme for molecular simulations. Given the paramount importance of the sampling problem to molecular simulations, any advance in this regard is extremely valuable to the community.

Stochastic resetting takes advantage of the fact that in many complex systems, the mean time required for an interesting transition to happen is larger than the median time. It is intuitive that, under these conditions, running several short MD trajectories will allow more transitions to be observed than running longer ones. The authors show that the rigorous formulation of SR allows to go beyond this simple argument, and even provide improvements to the well-established MetaD method. Of particular relevance is the fact that SR can bring significant improvements when it comes to estimating kinetic rates.

The paper is well written, and clearly presents the proposed method supporting it with several examples. It introduces a fresh perspective on the much-studied problem of enhanced sampling, which has the potential to advance the field and become relevant to the broad community of MD practitioners. It clearly deserves publication.

Response: Thank you for your very positive review, and for recommending publication.

The following are some minor points that if addressed could improve the present manuscript:

1. For completeness, the introduction should cite enhanced sampling methods that also rely on restarting trajectories, such as milestoning and weighted ensembles.

Response: We added the suggested citations on page 1 (refs. [9-13] of the revised manuscript): “Different methods have been developed to overcome this timescale problem, such as umbrella sampling, replica-exchange, free energy dynamics, milestoning, weighted ensemble, Metadynamics, On-the-fly probability enhanced sampling (OPES), and many others.”

2. A brief discussion should be devoted to how the choice of the starting point influences the COV and thus the speedup. Even better, a figure could be added for one of the 2D models, with a heatmap of the COV as a function of the starting coordinates.

Response: Analysis of the sensitivity of SR, as a standalone approach, to the initial configuration, can be found in the supporting information of our previous paper (<https://pubs.acs.org/doi/10.1021/acs.jpcllett.2c03055>.)

Following the reviewer's request, we added to the SI an analysis for SR combined with MetaD, and summarized the findings on page 3 of the main text: "Our results show minimal sensitivity to the initial positions. Plots similar to Figure 1c are given in the supporting information for additional initial positions, showing similar accelerations (Section 3)."

3. The term "speedup" can be ambiguous. It should be specified in the main text and/or in the figure caption that it refers to the frequency of transitions observed.

Response: To clarify, we moved the following definition of the speedup from the methods section to page 3: "The speedup is defined as the ratio of the mean number of timesteps before a first-passage is observed, between unbiased and biased simulations."

4. A discussion analogous to the following one should be added, if possible. This "speedup" does not in general correlate with the computational cost of estimating observables such as the free energy difference between two states. For example, a very aggressive non-tempered MetaD can lead to several transitions, but the resulting oscillating bias would make it impossible to estimate the free energy. Unless a good CV is used, such MetaD run would also not sample the correct transition state [16]. On the other hand, SR not only brings to more transitions per simulated time step, but it can in principle be used to estimate free energies. Furthermore, it does not alter the transition state, and in fact even allows to compute kinetic rates, as shown.

Response: We added this discussion to page 3: "We also acknowledge that very high speedups might come with a cost, e.g., very aggressive non-tempered MetaD can lead to several transitions, but the resulting oscillating bias would make it impossible to converge the desired properties. Resetting actually reduces this risk, by further accelerating the transitions while minimizing the bias deposition."

5. More emphasis could be placed on the fact that the proposed SR method is embarrassingly parallelizable, which is an extremely important characteristic.

Response: We thank the reviewer for this significant observation and emphasize it on page 10 of the revised manuscript in a comment that reads: "Resetting can be easily implemented in existing MD codes, and is highly parallelizable, as the trajectories between resetting events are entirely independent."

6. There is a typo in the SI, "longer then" -> "longer than"

Response: Thank you, we corrected it.

REVIEWERS' COMMENTS

Reviewer #1 (Remarks to the Author):

I thank the authors for implementing my suggestions and am excited to recommend this work for publication.

Reviewer #2 (Remarks to the Author):

Thanks to the authors for their clarifications - I found the additional details regarding the interplay between metaD and SR insightful - stimulating is also a possible role for steady-state dynamics, which I'm sure will resurface once this method will have established itself in the rare event community.

My concerns have been addressed, and following up on my initial assessment, I recommend publication of this manuscript in its present form.

Reviewer #3 (Remarks to the Author):

I am happy with the changes made to the manuscript, I think they improve clarity and make the proposed method more appealing..

I support publication.